# Copper-on-nitride enhances the stable electrosynthesis of multi-carbon products from $CO_2$

Zhi-Qin Liang[1,2], Tao-Tao Zhuang[1], Ali Seifitokaldani[1], Jun Li [1,3], Chun-Wei Huang[4], Chih-Shan Tan[1], Yi Li[5], Phil De Luna [6], Cao Thang Dinh[1], Yongfeng Hu[7], Qunfeng Xiao[7], Pei-Lun Hsieh[8], Yuhang Wang[1], Fengwang Li [1], Rafael Quintero-Bermudez[1], Yansong Zhou[1], Peining Chen[1], Yuanjie Pang[1,3], Shen-Chuan Lo[4], Lih-Juann Chen[8], Hairen Tan [1], Zheng Xu[2], Suling Zhao[2], David Sinton [3] & Edward H. Sargent [1]

Copper-based materials are promising electrocatalysts for $CO_2$ reduction. Prior studies show that the mixture of copper (I) and copper (0) at the catalyst surface enhances multi-carbon products from $CO_2$ reduction; however, the stable presence of copper (I) remains the subject of debate. Here we report a copper on copper (I) composite that stabilizes copper (I) during $CO_2$ reduction through the use of copper nitride as an underlying copper (I) species. We synthesize a copper-on-nitride catalyst that exhibits a Faradaic efficiency of 64 ± 2% for $C_{2+}$ products. We achieve a 40-fold enhancement in the ratio of $C_{2+}$ to the competing $CH_4$ compared to the case of pure copper. We further show that the copper-on-nitride catalyst performs stable $CO_2$ reduction over 30 h. Mechanistic studies suggest that the use of copper nitride contributes to reducing the CO dimerization energy barrier—a rate-limiting step in $CO_2$ reduction to multi-carbon products.

[1] Department of Electrical and Computer Engineering, University of Toronto, 10 King's College Road, Toronto, ON M5S 3G4, Canada. [2] Key Laboratory of Luminescence and Optical Information, Beijing Jiaotong University, Ministry of Education, Beijing 100044, China. [3] Department of Mechanical and Industrial Engineering, University of Toronto, 5 King's College Road, Toronto, ON M5S 3G8, Canada. [4] Material and Chemical Research Laboratories, Industrial Technology Research Institute, Hsinchu 31040, Taiwan. [5] Division of Nanomaterials & Chemistry, Hefei National Research Center for Physical Sciences at the Microscale, CAS Center for Excellence in Nanoscience, Hefei Science Center of CAS, Collaborative Innovation Center of Suzhou Nano Science and Technology, Department of Chemistry, University of Science and Technology of China, Hefei, Anhui 230026, China. [6] Department of Materials Science and Engineering, University of Toronto, 184 College Street, Toronto, ON M5S 3E4, Canada. [7] Canadian Light Source (CLS), 44 Innovation Boulevard, Saskatoon, SK S7N 2V3, Canada. [8] Department of Materials Science and Engineering, National Tsing Hua University, Hsinchu 30013, Taiwan. These authors contributed equally: Zhi-Qin Liang, Tao-Tao Zhuang, Ali Seifitokaldani.  Correspondence and requests for materials should be addressed to E.H.S. (email: ted.sargent@utoronto.ca)

Electrocatalytic $CO_2$ reduction has been investigated extensively based on metals such as Au, Ag, Sn, Zn, In, Pd, Cu, and their associated compounds[1–4]. Among these materials, Cu-based catalysts are promising for olefin and oxygenate production thanks to their moderate CO binding energies[5,6]. Multicarbon products such as ethylene ($C_2H_4$), ethanol ($C_2H_5OH$), and n-propanol ($C_3H_7OH$) are of great interest: $C_2H_4$, for example, is a valuable precursor in the manufacture of polymers[7]; $C_2H_5OH$ can be directly used as fuel[8]; and $C_3H_7OH$ has a higher mass energy density (30.94 kJ g$^{-1}$)[9,10] than does gasoline[11]. Furthermore, renewables-derived $C_2H_5OH$ and $C_3H_7OH$ can each be blended with gasoline to deliver a clean fuel[12].

Polycrystalline Cu metal is known to produce $CH_4$ with high selectivity[4,13], whereas oxide-derived Cu favors $C_{2+}$ products[14–17], a fact attributed to the effects of grain boundaries[18–20], high-local pH[21,22], and residual oxygen[14,23,24]. Certain prior computational studies have suggested that the $Cu^+/Cu^0$ mixture synergistically promotes $CO_2$ reduction to $C_{2+}$ products due to $CO_2$ activation and CO dimerization[25,26]. Experimentally, however, the stable presence of the active $Cu^+$ species during $CO_2$ reduction remains the subject of debate[27].

A $Cu^+$–$Cu^0$ core-shell structured catalyst offers an architecture wherein stable $Cu^0$ deposited on top of a $Cu^+$ support protects from further reduction. Recently, core-shell catalysts have been widely investigated in electrocatalysis and have achieved significantly improved activity and kinetics[28–35]. The core-support interactions modify the electronic structure of the surface catalyst, influencing the chemisorption of the intermediates in the electrocatalytic reaction[31]. Copper (I) oxide ($Cu_2O$), which has been mostly used as a precursor to Cu-based $CO_2$ reduction catalysts[14,17–19,23,24], is a candidate as a $Cu^+$ support; however, $Cu^+$ from $Cu_2O$ is unstable under $CO_2$ reduction conditions. Previous reports suggest that transition metal nitrides can be employed not only as a stable catalytic active species, but also as supports[36].

Here we sought therefore to investigate whether copper (I) nitride ($Cu_3N$) could be used as $Cu^+$ support during $CO_2$ reduction. We hypothesize that the $Cu_3N$ support affects the electronic structure and oxidation state of the surface Cu, decreasing the energy barrier associated with CO dimerization during $CO_2$ reduction. This, together with the prolonged presence of $Cu^+$ over time, could allow for the realization of increased-stability $C_{2+}$ electrosynthesis systems under $CO_2$ reduction conditions.

## Results

### Synthesis and structural characterization.
In order to challenge our hypothesis, we set out to synthesize Cu deposited on $Cu_3N$ (Cu-on-$Cu_3N$) catalyst as depicted in Fig. 1a. We first synthesized $Cu_3N$ nanocrystals capped with long-chain octadecylamine (ODA) ligands[37]. We then performed a ligand exchange using short-chain azide ($N_3^-$) to replace the ODA. An outer oxide was formed at the surface of $Cu_3N$ nanocrystals by exposing samples to ambient air during the ligand exchange process. These nanocrystals then went through an initial electroreduction process: we swept the cyclic voltammetry (CV) curve from 0 to −1.75 V vs. RHE to obtain the active Cu-on-$Cu_3N$ catalyst.

To investigate surface electronic properties, we conducted X-ray photoelectron spectroscopy (XPS) measurements of the samples (Fig. 1b). In the case of the $Cu_3N$ nanocrystals capped with ODA (Fig. 1b–i), the spectra of Cu 2p and Auger Cu LMM confirm a preponderance of $Cu^+$[38]. The sharp peak of N at a binding energy of 399 eV is consistent with that of the metal nitride[37,39]. Furthermore, X-ray diffraction (XRD) attests to the formation of $Cu_3N$ nanocrystals (Supplementary Fig. 1)[37].

Implementing the ligand exchange (Fig. 1b–ii and Supplementary Fig. 2) led to a different Cu composition compared to that before ligand exchange. A mixture of $Cu^{2+}$ and $Cu^+$ are present as observed in Cu 2p and LMM spectra[38], which suggests that copper (II) oxide (CuO) is formed in ambient air during the $Cu_3N$ ligand exchange. The new peak in the N 1s spectrum located at 403.1 eV aligns with that of the $N_3^-$ group in the ligand at the nanocrystal surface[40]. When taken together with Fourier-transform infrared (FTIR) spectra (Supplementary Fig. 3), these findings reveal that the ODA organic ligands are completely replaced by the $N_3^-$ short ligands. In addition, the Cu 2p peak areas indicate that the content of $Cu^{2+}$ is significantly higher relative to the $Cu^+$: we propose that CuO exists at the surface and substantially encompasses the $Cu_3N$. As shown in the O 1s spectrum (Supplementary Fig. 4), the dominant peak at 513.3 eV was assigned to O species in the surface CuO on the sample.

After initial reduction (Fig. 1b–iii), the Cu spectra show the presence of both $Cu^+$ and $Cu^0$, which indicates that the surface of the catalyst was reduced to Cu. The N 1s peak at 399 eV remains after reduction, indicating that the $Cu_3N$ phase is intact. The disappearance of the N peak at 403.1 eV, which is the characteristic of the $N_3^-$ ligands, can be ascribed to the weak electrostatic interaction between the ligands and the surface of the $Cu_3N$ nanocrystals when a potential was applied.

We used transmission electron microscopy (TEM) to investigate further the structure of the catalyst (Supplementary Fig. 5). Before ligand exchange, $Cu_3N$ nanocrystals have an average diameter of 30 nm. After ligand exchange, a reduced spacing between the nanocrystals is observed, similar to the case of quantum dot ligand exchanges[41].

The local atomic-scale elemental composition on individual Cu-on-$Cu_3N$ nanoparticle was further examined (Fig. 2a, b). From high-resolution transmission electron microscopy with electron energy loss spectroscopy (HRTEM-EELS, Fig. 2c), we observed that Cu was distributed across the volume of each nanoparticle; while N was concentrated in the core and was notably lower at the surface. The catalyst surface exhibited indications of surface reconstruction following operation under reducing conditions[42]. Our analysis of HRTEM-EELS data indicates $a <= 3$ nm surface Cu layer on top of $Cu_3N$ (Supplementary Fig. 6).

We further investigated the distribution of N using EELS spectra. In a given nanoparticle, looking at two different positions (Fig. 2d), we found that for point A (inner), a strong absorption feature starting from 401 eV was obtained, consistent with the N K-shell absorption edge[43]. No obvious absorption signal was observed for point B (surface), indicating no N at the surface. These observations indicate a Cu-on-$Cu_3N$ structure.

We synthesized Cu deposited on $Cu_2O$ (Cu-on-$Cu_2O$) and pure Cu catalysts as control samples using a process similar to the synthesis of the Cu-on-$Cu_3N$ catalyst. XRD patterns confirm the formation of Cu-on-$Cu_2O$ and pure Cu after electroreduction (Supplementary Fig. 7a–c). Their morphology and size are similar to those of the Cu-on-$Cu_3N$ catalyst (Supplementary Fig. 7b–d). Double-layer capacitance measurements yielded electrochemical roughness factors of 9.7, 8.0, and 9.3 for the Cu-on-$Cu_3N$, Cu-on-$Cu_2O$, and Cu catalysts, respectively, indicating similar surface: geometric area ratios (Supplementary Fig. 8 and Supplementary Table 1).

We also obtained valence band spectra (VBS) to examine differences in the valence electronic structure between the Cu and the composite (Supplementary Fig. 9). Comparing with the case of pure Cu, we found that the Fermi-level ($E_F$) shifted toward the $VB_m$ by 0.33 eV for Cu-on-$Cu_3N$ and 0.08 eV for Cu-on-$Cu_2O$, respectively, indicating that the core-level $Cu_3N$ and Cu2O supports have an effect on the electronic structure of the surface Cu.

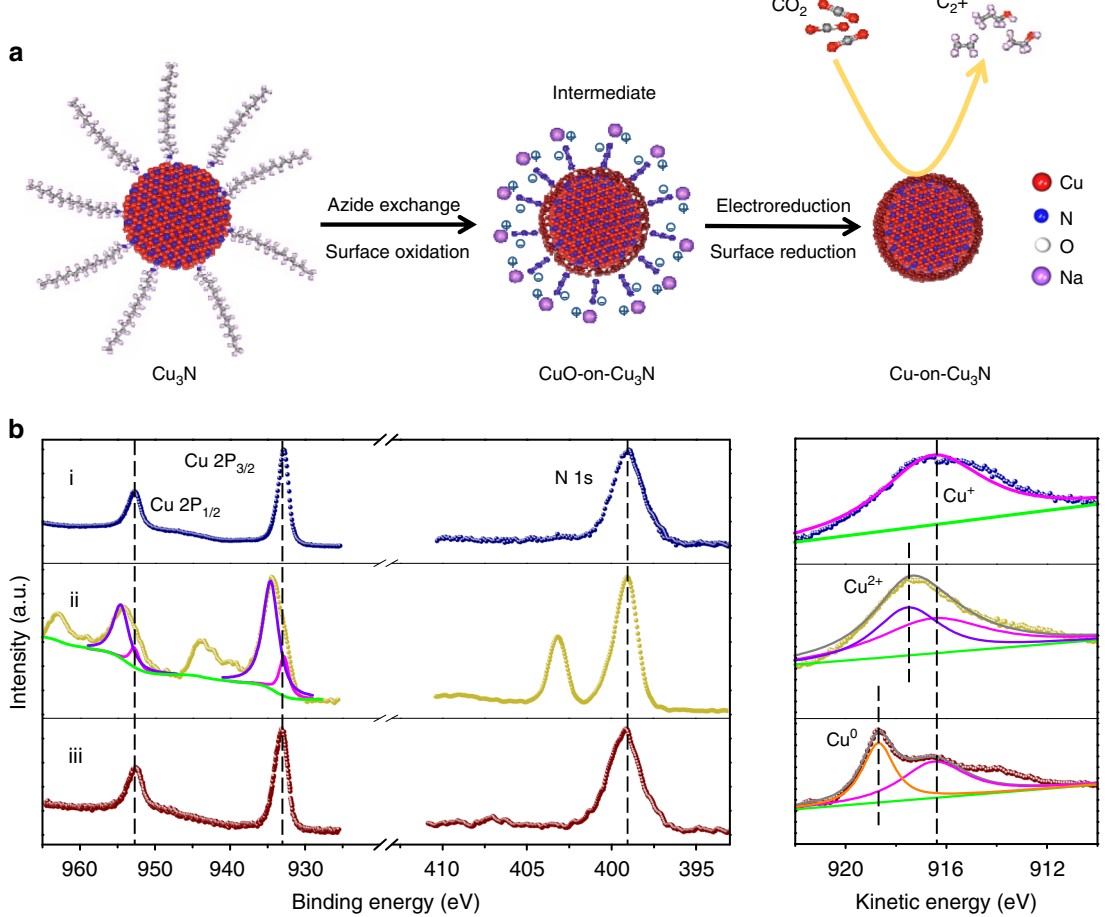

**Fig. 1** Electrocatalyst design and the corresponding XPS characterization. **a** Schematic of preparing the Cu-on-Cu₃N catalyst. **b** XPS spectra of Cu 2p, N 1 s, and Auger Cu LMM of the Cu₃N nanocrystals with long organic ODA (i), the Cu₃N nanocrystals with an oxide layer after N₃⁻ ligand exchange (ii), and the Cu-on-Cu₃N composite after initial electroreduction (iii)

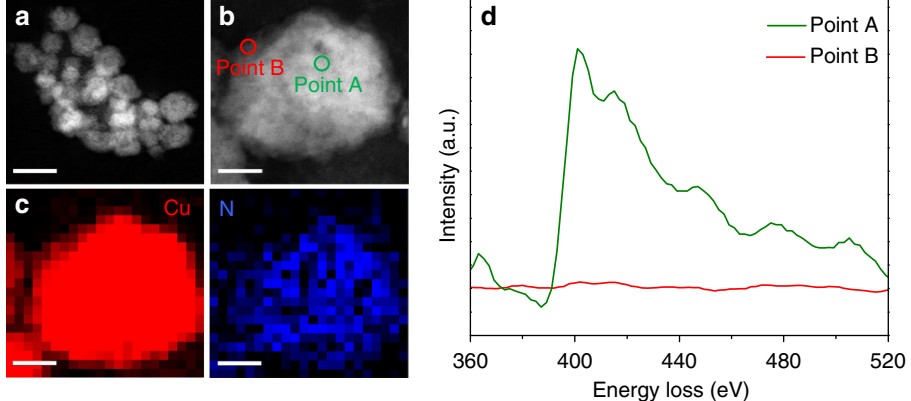

**Fig. 2** TEM characterization of the Cu-on-Cu₃N catalyst. **a, b** HADDF-STEM images. **c** STEM-EELS Cu and N Element mapping of one individual particle in **b, d**, EEL spectra of element N K-edge circled as point A and point B in **b**. The scale bars are 50 nm in **a**, and 10 nm in **b** and **c**

**Spectroscopic characterization**. To investigate the structure and chemical state of the active catalysts with time evolution under $CO_2$ reduction, we obtained in situ X-ray absorption spectra (XAS) of the three catalysts at −0.95 V vs. RHE during $CO_2$ reduction.

As depicted in Fig. 3a, b, the Cu K-edge XAS spectrum of the as-prepared Cu-on-Cu₃N catalyst presents an absorption edge between Cu (8979 eV) and Cu₃N (8980.5 eV)—and in particular exhibits a prominent shoulder at 8980.0 eV. Over

the course of $CO_2$ reduction, both Cu and Cu₃N features are still present, with a shoulder energy at 8979.4 eV after 2 h. In contrast, the Cu-on-Cu₂O catalyst shows a prominent metallic Cu feature after 1 h (Supplementary Fig. 10a). Pure Cu presents a metallic Cu feature under $CO_2$ reduction (Supplementary Fig. 10b).

To gain more insight into the role of the Cu⁺ support, we acquired in situ Cu K-edge spectra of Cu-on-Cu₃N and Cu-on-Cu₂O catalysts following 30 min under $CO_2$ reduction (Fig. 3c).

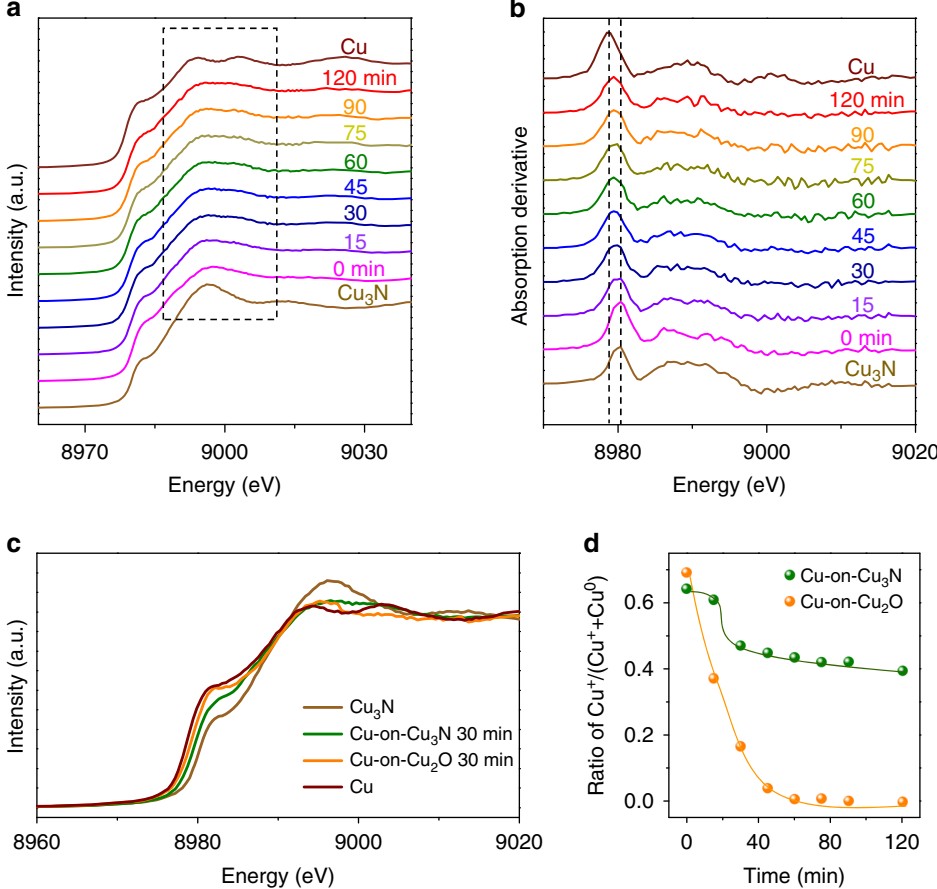

**Fig. 3** In situ characterization of the structure and chemical state for the catalysts during $CO_2$ reduction. **a** Cu K-edge XAS spectra of the Cu-on-$Cu_3N$ catalyst as function of reaction time at −0.95 V vs RHE. **b** The first derivatives of the spectra in **a**. **c** In situ Cu K-edge spectra during the initial 30 min on the catalysts: Cu-on-$Cu_3N$ (green) and Cu-on-$Cu_2O$ (orange). Spectra of Cu (red) and $Cu_3N$ (yellow) are also listed as references. **d** Ratio of $Cu^+$ relative to the reaction time at −0.95 V vs RHE

We found that the absorption edges of the two catalysts are between $Cu^+$ and $Cu^0$, indicating the presence of a mixture during the reaction. However, the absorption edge of Cu-on-$Cu_2O$ was at a lower energy than that of Cu-on-$Cu_3N$, with energies at 8979.4 eV and 8979.8 eV, respectively. We also calculated the ratio of Cu oxidation states as function of the reaction time (Fig. 3d). The Cu-on-$Cu_3N$ catalyst shows that the structure becomes stable with $Cu_3N$ and Cu after the initial 60 min, while Cu-on-$Cu_2O$ only presents the Cu component after 1 h. This observation indicates suppression of the partial reduction of the catalyst when we use the $Cu_3N$ support.

We sought a method to probe with greater surface-specificity catalyst as function of reaction time. We acquired angle-resolved XPS (ARXPS) at a 20° emission angle relative to the sample normal (Supplementary Fig. 11a). The detection depth is below 2 nm[44]. Cu LMM spectra (Supplementary Fig. 11b–f, left column) indicate the presence of $Cu^+$ and $Cu^0$, and the N 1 s spectra (Supplementary Fig. 11 b–f, right column) are consistent with the spectrum of metal nitride, indicating the presence of $Cu_3N$ in the first ~2 nm of the surface over the course of $CO_2$ reduction. In the initial 60 min, $Cu^0$ content increased and $Cu_3N$ content decreased; thereafter, such as following a 2-h reaction, the catalyst gradually reached a stable surface composition. This result agrees with the observed in situ XAS data (Fig. 3d).

Both in situ XAS and ex situ ARXPS indicate the presence of $Cu^+$ following $CO_2$ reduction. Further, the N signal suggests the presence of $Cu_3N$, and STEM-EELS mapping (Supplementary Fig. 6) shows evidence of $Cu_3N$ in the subsurface layer following

$CO_2$ reduction. Nevertheless, we point out also that XAS has a bulk penetration depth; and that air-sensitive Cu complicates the interpretation of the ARXPS studies herein. For these reasons, direct and unambiguous confirmation of the presence of $Cu^+$ at the surface of the catalyst remains an ongoing opportunity for further advances in the field of Cu-based electrocatalysis and model development.

**$CO_2$ electroreduction performance**. To verify the effect of the $Cu^+$ support on the surface catalyst, we carried out $CO_2$ reduction using the Cu-on-$Cu_3N$, Cu-on-$Cu_2O$, and pure Cu catalysts, respectively. To analyze the selectivity toward various products with different applied potentials, we performed stepped-potential electrolysis between −0.55 and −1.45 V vs RHE (with *iR* correction in Supplementary Fig. 12).

Cu-on-$Cu_3N$ gives the highest $C_{2+}$ production among the three catalysts (Fig. 4a, b). When the applied potential is less negative than −0.65 V vs RHE, $CH_4$, and HCOOH are the main products; whereas, when we sweep toward more strongly negative potentials, we obtain production of reduced $C_{2+}$ species, such as $C_2H_4$, $C_2H_5OH$, and $C_3H_7OH$. This indicates CO dimerization beyond the potential of −0.65 V vs. RHE (Supplementary Figs. 13a, 14 and Supplementary Table 2). The highest FE for total $C_{2+}$ reaches 64 ± 2% at −0.95 V vs. RHE, with $C_2H_4$, $C_2H_5OH$, and $C_3H_7OH$ accounting for 39 ± 2%, 19 ± 1%, and 6 ± 1%, respectively.

The Cu-on-$Cu_3N$ catalyst achieves a 6.3-fold enhancement in the ratio of $C_{2+}$ to $CH_4$ compared to Cu-on-$Cu_2O$; and a 40-fold

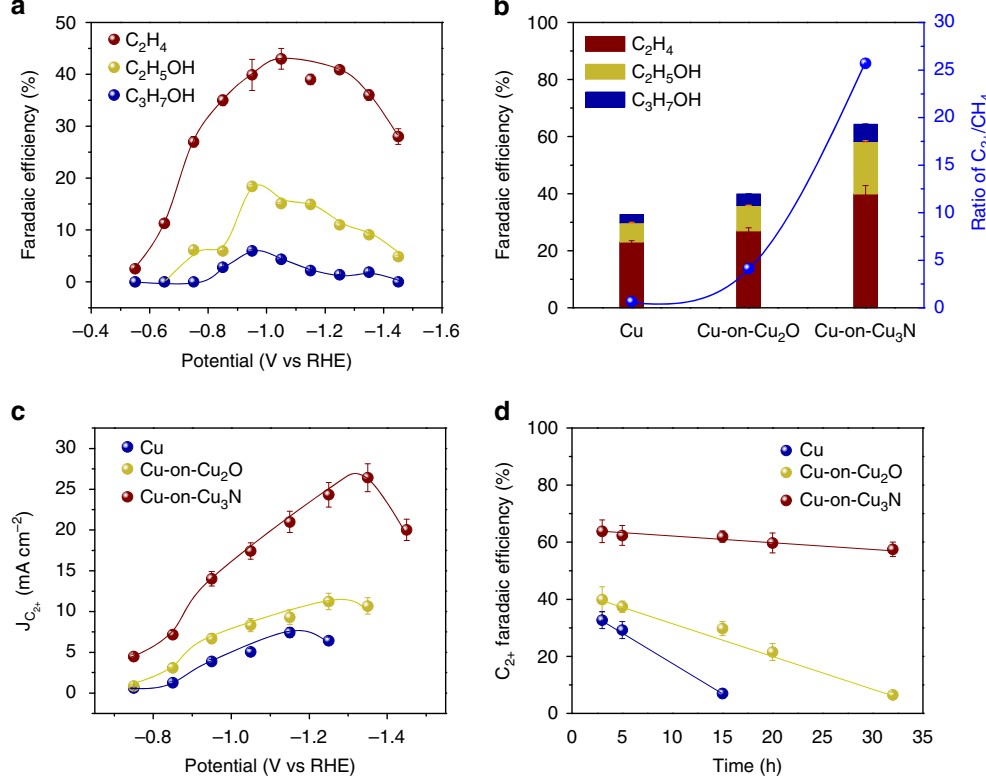

**Fig. 4** CO$_2$ electroreduction performance of the designed catalysts. **a** Faradaic efficiency of the C$_{2+}$ distribution on Cu-on-Cu$_3$N at different potentials. **b** Comparison of faradaic efficiency for C$_{2+}$ and the ratio of C$_{2+}$/CH$_4$ at −0.95 V vs RHE on Cu, Cu-on-Cu$_2$O, and Cu-on-Cu$_3$N. **c** C$_{2+}$ partial current density at different potentials on the three catalysts. **d** Stability test of C$_{2+}$ selectivity on the three catalysts. Experiments from **a** to **d** were performed in triplicates and the results are shown as mean ± standard deviation

enhancement over pure Cu (Fig. 4b). CH$_4$ production is thus strongly suppressed for the catalysts that use Cu$_2$O and Cu$_3$N support compared with pure Cu (Supplementary Fig. 13b–c and Supplementary Tables 3−4).

To further compare the CO$_2$ reduction activity of the three samples, we obtained C$_{2+}$ partial current densities at a variety of potentials between −0.75 and −1.45 V vs RHE (Fig. 4c). The Cu-on-Cu$_3$N catalyst exhibits a higher C$_{2+}$ partial current density relative to Cu-on-Cu$_2$O and pure Cu across the entire potential window, with a maximum 14 mA cm$^{-2}$ at −0.95 V vs RHE, 2.2x and 4.4x higher than Cu-on-Cu$_2$O and pure Cu catalysts, respectively.

To test the operational stability of the catalysts, we carried out CO$_2$ reduction over an extended period of time. The Cu-on-Cu$_3$N catalyst exhibits relatively stable Faradaic efficiencies toward C$_2$H$_4$, C$_2$H$_5$OH, and C$_3$H$_7$OH, with a relative 10% decrease following 30 h of continuous CO$_2$ electroreduction (Fig. 4d and Supplementary Fig. 15). We attribute this superior stability to the suppressed reduction of the Cu$_3$N support over time, such that the beneficial effect of the Cu$_3$N support is sustained over this longer operating time. In contrast, Cu-on-Cu$_2$O catalyst shows a loss of about 25% relative of its selectivity following 15 h, while pure Cu shows decreased C$_{2+}$ production following 5 h of CO$_2$ reduction (Fig. 4d).

## Discussion

Recent studies have suggested that the CO$_2$ reduction performance of oxide-derived Cu catalysts can be ascribed to local pH and to derived surface defects[19,21,22]. Compared to pure Cu, Cu-on-Cu$^+$ catalysts display a suppression in methane selectivity, which can be attributed to increased local pH. Comparing Cu-on-Cu$_2$O and Cu-on-Cu$_3$N catalysts, the geometric current densities are similar

(Supplementary Fig. 16), which indicates a nearly identical consumption rate of local protons during CO$_2$ reduction. We propose therefore that differences in local pH do not account for the higher C$_{2+}$ selectivity for Cu-on-Cu$_3$N relative to Cu-on-Cu$_2$O.

We also considered surface defects as another possible contributing mechanism. For the Cu-on-Cu$_2$O catalyst, we believe that surface defects—grain boundaries—may influence the selectivity toward C$_{2+}$ in the case of the oxide-derived process. For the Cu-on-Cu$_3$N catalyst, surface defects can also affect the C$_{2+}$ selectivity. However, compared with Cu-on-Cu$_2$O, which was quickly derived to Cu (Fig. 3d), the Cu-on-Cu$_3$N catalyst retained a higher C$_{2+}$ selectivity under CO$_2$ reduction. We offer that suppressed reduction of the Cu$_3$N support thus plays a significant part in the high selectivity over the course of CO$_2$ reduction.

Single-crystal studies have also shown that the exposed Cu facets affect selectivity[45–47]. In this work, we synthesized three catalysts using an initial electroreduction of the surface oxidation layer using a negative cyclic voltammetry (CV) scan. During this process, the Cu species possess a polycrystalline structure (Supplementary Fig. 17). These structures do not exhibit the specific facet orientation. Therefore, we would not expect that these would contribute in a quantitatively significant way to increase C$_{2+}$ selectivity.

Taking these findings together with those from XPS (including ARXPS and VBS), in situ XAS, and HRTEM-EELS, we propose that the Cu$^+$ support may play a role in selectivity toward C$_{2+}$. Due to the change of surface structure with time evolution, the surface Cu layer is no longer uniform and some of Cu$^+$ may reside in the subsurface layer during the initial reduction (Supplementary Fig. 6), favouring selectivity for C$_{2+}$. Cu$_3$N as the support stabilizes the Cu$^+$ to a greater degree than does Cu$_2$O during CO$_2$ reduction (Fig. 3d), leading to heightened C$_{2+}$ production.

To understand the role of $Cu^+$ support in $Cu^0$-on-$Cu^+$ composite catalyst for $CO_2$ reduction, we performed density functional theory (DFT) computations to calculate the oxidation state of the surface Cu in the models: Cu, $Cu_3N$, $Cu_2O$, Cu-on-$Cu_3N$, and Cu-on-$Cu_2O$ (Supplementary Figs. 18–22). Bader charge analyses show that both $Cu_2O$ and $Cu_3N$ as $Cu^+$ supports modulate the partial oxidation state of the surface copper layer (Supplementary Table 5), with +0.03 extra charge induced by $Cu_2O$ support and +0.25 by $Cu_3N$ support on (100) facet, respectively, different from that of pure Cu (0), pure $Cu_2O$ (+0.26), and pure $Cu_3N$ (+0.47) on (100) facet. This modulated partial oxidation state enables Cu-on-$Cu_3N$ to achieve the lowest CO dimerization barrier energy (0.884 eV) among all models (Supplementary Figs. 23–24 and Supplementary Table 5), thereby indicating promise as a candidate for $C_{2+}$ production. To evaluate further the selectivity of these catalysts for $C_{2+}$ products compared to the competing $C_1$ products, we also calculated the energy barriers for CO protonation (Supplementary Methods). The results reveal that the energy barrier for the $C_1$ pathway for Cu-on-$Cu_3N$ (0.933 eV) is higher than that of Cu-on-$Cu_2O$ (0.749 eV) and pure Cu (0.721 eV) on (100) facets (Supplementary Table 5).

Since, the stability of the sublayer $Cu^+$ is important, we studied the diffusion free energy barrier for nitrogen and oxygen from their original positions in the $Cu_3N$ and $Cu_2O$ structures, respectively, to the surface of Cu-on-$Cu_3N$ and Cu-on-$Cu_2O$ with 4 Cu top layers. Although there is a large energy barrier (>2 eV) for both O and N to leave their original position, there is no more diffusion barrier for O from the first layer to the surface (Supplementary Figs. 25–26 and Supplementary Table 6). However, for N we observe another energy barrier (~1 eV) to diffuse to the surface (Supplementary Fig. 27 and Supplementary Table 6). This agrees with our observations throughout that N in the sublayer is more stable than O.

In summary, the present work introduces a Cu-based catalyst that enables metallic Cu-on-$Cu_3N$ to promote the production of $C_{2+}$ species. $Cu_3N$ was chosen as the inner support to modify the electronic structure of the surface metal, affecting thereby the adsorption and dimerization of intermediate CO properly in the $CO_2$ reduction. Together with the suppressed reduction of $Cu^+$ using $Cu_3N$ as the support, we were able to achieve higher selectivity for $C_{2+}$ formation using Cu-on-$Cu_3N$ compared to the case of Cu-on-$Cu_2O$ and pure Cu.

## Methods

**Synthesis of $Cu_3N$ nanocrystals.** Quantity of 0.15 g of $Cu(NO_3)_2 \cdot 3H_2O$ and 4.3 g of 1-octadecylamine (ODA) was dissolved in 15 mL of 1-octadecene. The solution was degassed for 10 min at 150 °C. The temperature was then raised to 240 °C and kept for another 10 min. When it cooled down to room temperature, the product was collected by centrifugation, washed with hexane/acetone (1/4) three times, and finally dispersed in hexane.

**Transformation of $Cu_3N$ to CuO-on-$Cu_3N$.** We used the ligand exchange method in ambient air to achieve the transformation of $Cu_3N$ to CuO-on-$Cu_3N$. Ten milligrams of $Cu_3N$ nanocrystals with organic ligands was dissolved in 1 mL of hexane (10 mg $L^{-1}$), while 10 mg of sodium azide ($NaN_3$) was dissolved in 1 mL of NMF (10 mg $L^{-1}$). The two solutions were then mixed and stirred overnight. The nanocrystals gradually transferred to NMF. The bottom phase was extracted and washed with hexane three times. The $N_3^-$-capped $Cu_3N$ nanocrystals were then precipitated out using chloroform as the anti-solvent. The precipitate was dried in vacuum for 15 min and then stored. In the ligand exchange process, we deliberately exposed the materials to ambient air to introduce an oxide layer at the surface of $Cu_3N$ nanocrystals.

**Transformation of CuO-on-$Cu_3N$ to Cu-on-$Cu_3N$.** We conducted the initial electroreduction for the CuO-on-$Cu_3N$ sample by sweeping the cyclic voltammetry (CV) curve from 0 to −1.75 V vs RHE at a rate of 50 mV $s^{-1}$, yielding the Cu-on-$Cu_3N$ catalyst.

**Synthesis of control Cu-on-$Cu_2O$ and pure Cu catalysts.** $Cu_2O$ and Cu nanocrystals were synthesized using 0.5 g of $Cu(NO_3)_2 \cdot 3H_2O$ and 0.05 g of Cu

$(NO_3)_2 \cdot 3H_2O$ instead, respectively, while keeping other experimental conditions the same as in the synthesis of $Cu_3N$ nanocrystals. The ligand exchange and initial electroreduction processes were the same as in the case of the Cu-on-$Cu_3N$ catalyst.

**Working electrode preparation.** Ten milligrams of the catalyst was dispersed in 1 mL of methanol, including with 20 μL of Nafion solution (anhydrous, 5 wt %) by sonicating for 30 min. Twenty microliter of the homogeneous solution was then loaded on a glassy carbon electrode. The geometric surface area was 0.19 $cm^2$. The electrode was dried in ambient air for the subsequent $CO_2$ electroreduction test.

**Electrochemical measurement.** Electrochemical tests were performed in a two-compartment H-cell. A proton exchange membrane (Nafion 117) was used. The electrolyte was 30 mL of 0.1 M $KHCO_3$ solution saturated with $CO_2$ gas in the cathode part for at least 30 min prior to the $CO_2$ reduction test. Platinum was used as the counter electrode and Ag/AgCl as the reference electrode (saturated with 3.0 M KCl, BASi). The glassy carbon electrode loaded with the catalyst served as the working electrode. Liner sweep voltammetry (LSV) with a scan rate of 50 mV/s was conducted first. The gas products were detected using a gas chromatograph (GC, PerkinElmer Clarus 600) equipped with a thermal conductivity detector (TCD) for hydrogen ($H_2$) quantification and a flame ionization detector (FID) for methane ($CH_4$) and ethylene ($C_2H_4$). Liquid products were quantified using $^1H$ nuclear magnetic resonance (NMR, Agilent DD2 500). The NMR samples were prepared by mixing 0.5 mL of electrolyte with 0.1 mL of deuterated water ($D_2O$), and 0.02 μL of dimethyl sulfoxide (DMSO) was added as an internal standard. Potential $E$ was converted to the RHE reference electrode using:

$$E \, (\text{versus RHE}) = E \, (\text{versus Ag/AgCl}) + 0.197 \, \text{V} + 0.059 \, \text{V} \times \text{pH}.$$

**Electrochemical active surface area (ECSA) measurement.** We used the double layer capacitance method to measure the surface roughness factors ($R_f$) for the samples relative to polycrystalline Cu ($R_f = 1$) foil. $R_f$ was calculated from the ratio of the double-layer capacitance C of the catalyst electrode and Cu foil electrode ($C_{Cu \, foil} = 29$ μF), that is, $R_f = C/C_{Cu \, foil}$. C was determined by measuring the geometric current density at a potential at which no Faradaic process was occurring when we varied the scan rate of the CV. CV was performed in the same electrochemical cell with 0.1 M $KHCO_3$ electrolyte separated with a Nafion proton exchange membrane. The linear slope provides C. ECSA = $R_f \times S$, where S stands for the geometric area of the glassy carbon electrode ($S = 0.19 \, cm^2$ in this work).

**Characterization.** XRD were measured on a Philips X'Pert Pro Super X-ray diffractometer equipped with graphite-monochromatized Cu Ka radiation. X-ray photoelectron spectroscopy (XPS) was carried out on an ESCA Lab MKII X-ray photoelectron spectrometer. The source for excitation is Mg Ka radiation. For angle-resolved XPS (ARXPS), the samples were fixed on a rotatable holder, which enables measurement for take-off angles θ of 20° measured relative to the surface normal. Low-resolution transmission electron microscopy (TEM) studies were performed on JEOL-2010F with an acceleration voltage of 200 kV. High-angle annular dark field scanning transmission electron microscopy (HAADF-STEM) and high-resolution transmission electron microscope electron energy loss spectroscopy (HRTEM-EELS) were carried out using a cold-field emission Cs-corrected JEOL ARM-200F Atomic Resolution Analytical Microscope operating at an accelerating voltage of 200 kV. In situ X-ray absorption of the Cu K-edges was performed at the Soft X-ray Microcharacterization Beamline (SXRMB) at Canadian Light Source (CLS). A homemade in situ electrochemical cell was used, with platinum as the counter electrode and Ag/AgCl as the reference electrode. The electrolyte is $CO_2$-purged 0.1 M $KHCO_3$. The acquisition of each spectrum took 15 min.

## Data availability

The data that support the findings of this study are available within the article and its Supplementary Information files. All other relevant source data are available from the corresponding author upon reasonable request.

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

## Acknowledgements

This work was supported financially by the Ontario Research Fund Research-Excellence Program, the Natural Sciences and Engineering Research Council (NSERC) of Canada, and the CIFAR Bio-Inspired Solar Energy program. Computations were performed on the SOSCIP Consortium's Blue Gene/Q computing platform. SOSCIP is funded by the Federal Economic Development Agency of Southern Ontario, the Province of Ontario, IBM Canada Ltd., Ontario Centres of Excellence, Mitacs and 15 Ontario academic member institutions. X-ray absorption spectra were performed on SXRMB beamlines at the Canadian Light Source (CLS), which is supported by the Canada Foundation for Innovation, Natural Sciences and Engineering Research Council of Canada, the University of Saskatchewan, the Government of Saskatchewan, Western Economic Diversification Canada, the National Research Council Canada, and the Canadian Institutes of Health Research. Z.L. acknowledges a scholarship from the China Scholarship Council (CSC) (201607090041) and Basic and Innovation Program, Beijing Jiaotong University (KSJB17016536). A.S. acknowledges Fonds de Recherche du Quebec-Nature et Technologies (FRQNT) for the postdoctoral award. P.D.L. acknowledges support from NSERC in the form of the Canada Graduate Scholarship. H.T. acknowledges the Netherlands Organisation for Scientific Research (NWO) for a Rubicon grant (680-50-1511) to support his postdoctoral research at University of Toronto. The authors thank C. Q. Zou, M. X. Liu, F. F. Fan, J. Xing and L. Gao from University of Toronto for fruitful discussions.

## Author contributions

E.H.S. supervised the project. Z.L. and T.Z. conceived the ideas, conducted the experiments, analyzed the results, and wrote the manuscript. A.S. carried out simulations. C.H. and P.H. conducted the TEM measurements. J.L., Y.H. and Q.X. assisted in measuring XAS and analyzing the results. S.L., L.C. and C.T. assisted in analyzing the TEM results. R.Q.-B. and Y.L. performed XPS measurements. Y.Z. assisted in analyzing the XPS results. Y.W. and F.L. assisted in discussing mechanisms. P.D.L., C.T., H.T. and Y.P. assisted in revising the

manuscript. P.C., Z.X., S.Z. and D.S. assisted in reviewing the manuscript. All authors read and commented on the manuscript.

## Additional information

**Competing interests:** The authors declare no competing interests.

