## [Peer Review File · Nature Communications]

Reviewers' comments:

Reviewer #1 (Remarks to the Author):

Contents of manuscript: Authors prepared Cu on Cu₃N, which they believe to be an effective catalyst for producing C₂ products from CO₂. The premise for this move is that the Cu₃N core helps to keep the outer layer of Cu oxidized. The highest CO₂R activities found are at -0.95 V RHE: C₂H₄ (39% FE), C₂H₅OH (19% FE) and propanol (6% FE).

There are some issues with this manuscript:

1. No operando or in situ characterization techniques were done to ensure that the Cu outer layer stays oxidized during CO₂ reduction. Hence, the foundational evidence to support the basic claims of this manuscript is not there.
2. The CO₂R activities reported above are suppose to be the best. But really, they are not any better than many previous works using the so-called Cu₂O-derived Cu.
3. Minor: Please include error bars in data.
4. The authors has from the very start, make the statement that Cu⁺ are active sites, and has to be protected from reduction. This claim is not that well supported in the CO₂R community. Could the authors give a more balanced viewpoint that metallic Cu sites with defects could also work for CO₂R - > C₂+. Moreover, how about the roles of local pH, as proposed by Koper's group?

Because of # 1, #2 and #4, I am not able to recommend acceptance of this manuscript, in its present form. Much more work needs to be done, especially to address #1.

Reviewer #2 (Remarks to the Author):

This work is a combined theory and experiment study that examines the effect of oxidized Cu supports on the activity, selectivity, and stability of thin Cu overlayers. Overall, the authors conclude that a lower kinetic barrier for C-C coupling in the case of Cu overlayers on both Cu₃N and Cu₂O supports leads to an increase in activity for CO₂ reduction to C₂⁺ products. The impact of sub-surface oxygen on the catalytic behavior of Cu-based catalysts has been a widely-debated topic in the CO₂ electrochemistry field, and developing a better fundamental understanding on the structure and stability of active-site motifs in such composite catalysts is of general interest to the scientific community. While the performance of the composite Cu-based catalysts in this study are very impressive, the correlation between structure and activity is weak and will likely require additional experiments to substantiate the conclusions made by the authors. As such, I cannot recommend that this work is published in Nature Communications in its current state, and have outlined my concerns below.

- According to a recent theoretical study by Garza, Bell, and Head-Gordon, the diffusion rates of anions such as oxygen in an oxidized Cu lattice are expected to be quite fast, and thus, sub-surface Cu oxides are expected to have poor stability within the atomically thin overlayer structures suggested by the authors (DOI: 10.1021/acs.jpcclett.7b03180). The partially reduced Cu₂O and Cu₃N structures generated by DFT have oxygen or nitrogen within the first several monolayers of the surface. In the cases where metallic Cu is an overlayer on Cu₂O or Cu₃N, what is the expected stability of the oxidized structures in either the 3rd or 2nd monolayer, respectively.

- The geometric arrangement of Cu atoms within the metallic overlayers on Cu₂O and Cu₃N appear to be quite different. Previous work in single crystal electrochemistry has shown that the C-C coupling activity of Cu electrodes are quite sensitive to the exposed facet. Are these the most thermodynamically stable facet terminations for the metallic overlayer? Otherwise, why were these particular facets chosen? There doesn't seem to be a strong correlation between the CO adsorption energies and the kinetic barriers for CO dimerization. How much of the decrease in the kinetic barrier for CO dimerization is due to a geometric effect vs. a ligand or strain effect from the support?
- In the best case scenarios of having the metallic overlayers on Cu₂O and Cu₃N, the improvements in CO dimerization barrier height appear to be quite small in comparison to metallic Cu (~0.1 eV). In particular, the differences in barrier height between the metallic overlayers on Cu₂O and Cu₃N appear to be very small. Are these improvements within the error of the DFT calculations?
- For the Cu₂O or Cu₃N support to affect the binding of reaction intermediates through either ligand or strain effects, the metallic Cu overlayer must be very thin and the authors suggest a thickness of ~1-2 monolayers. However, the combination of physical characterization techniques utilized by the authors does not have a high enough spatial resolution to determine how thick this metallic Cu overlayer is on the Cu₂O or Cu₃N support. Experimentally demonstrating either the thickness of the overlayer or that its electronic structure is affected by the Cu₂O or Cu₃N support is critical to substantiate the enhancement mechanism that the authors have proposed. Have the authors considered valence band spectroscopy measurements to examine the differences in electronic structure between the metallic Cu nanoparticles and composite structures?
- Instead of a strain or ligand effect from the Cu₂O or Cu₃N support, have the authors considered a mechanism of enhancement from a difference in exposed Cu facets? Single-crystal studies have shown similar trends in C₂+ / C₁ ratios through improvements in C-C coupling activity (and CH₄ suppression) from introducing more undercoordinated sites on the surface (DOI: 10.1007/978-0-387-49489-0_3).
- How does the stoichiometry of the Cu overlayer on Cu₂O or Cu₃N composite change after steady state electrocatalysis measurements? Are there any bulk structural changes in the catalyst after these longer-term measurements? Is there any evidence of O₂- or N₃- leaching over time?
- Why is the Cu overlayer on Cu₃N or Cu₂O more stable than pure Cu? The transmission electron micrographs in the Supporting Information don't show much change in the particle morphology after the reduction treatment. Have the authors used ICP measurements to determine whether there is any dissolution or delamination of Cu from the electrodes over time?
- Please check the fitting of the X-ray photoelectron spectra. In particular, the fitting for the Cu Auger LMM spectra needs more attention to be published.
- It is difficult to deconvolute the effect of the ligand from the catalytic behavior of Cu₃N for the measurements shown in Figure S13 and Table S6.

Reviewer #3 (Remarks to the Author):

This study explores the design of Cu-based catalysts where Cu metal (Cu⁰) is deposited on a Cu⁺ support to enhance the selectivity of CO₂ reduction to C₂+ products. The idea is partly supported by literature work on Cu oxide derived catalysts and DFT calculations within the paper evaluating CO

adsorption energy versus CO dimerization barrier. The hypothesis is that the selectivity step between C1 and C2 products is hydrogenation of CO versus CO dimerization. Based on the finding that Cu on Cu₃N shows that the CO dimerization barrier is lowered in comparison to Cu metal and Cu on Cu₂O, they synthesis Cu-on-Cu₃N catalysts. These catalysts were tested along with Cu and Cu-on-Cu₂O catalysts. They see that the FE to C2 is enhanced substantially on the Cu-on-Cu₃N catalysts. Very interestingly (and this is rare in the CO₂ electroreduction work in the literature) they report the stability of the CO₂ electroreduction over 32 hours and see much more stability than Cu-on-Cu₂O and Cu metal. This stability result is promising since one of the issues with much of the CO₂ electroreduction work is that the performance of the catalysts degrades rapidly.

Overall, this paper reports a very interesting new finding for CO₂ electroreduction. This is a hot area with a lot of results coming quickly in the past 2-3 years but this paper is promising since they not only see a dramatic increase in selectivity but also study stability and have a hypothesis-driven approach to the catalyst design. I am not entirely convinced their hypothesis is correct for the reasons they give – i.e enhance CO dimerization. The main suggestion and question I have are related to the DFT calculations, which I found to be relatively weak component of this paper.

(1) They should report more detail on the CO dimerization calculation. There has been some controversy about this pathway – the initial proposal was that a CO + CO⁻ reaction occurs. Not two neutral CO dimers, which was found to have a very large barrier. The more recent work from the Norskov group examines this reaction and showed that cations and protons could be used to facilitate this reaction. There are no references to the work of Koper and Norskov (and also Goddard on the CO dimerization on Cu(100)) in this paper. I realize that references are limited but the authors should provide this context in the SI and explain what was involved in their CO dimerization calculation in reference to this work. If two neutral CO molecules were coupled to form a CO dimer then this would be a non-electrochemical step and would not explain any of the potential dependence seen in the experiments.

(2) If the hypothesis is that C1 versus C2 is a competition between CO hydrogenation and CO dimerization, does it not make sense to look at the barrier to CO hydrogenation on these various surfaces? i.e. CO* + H⁺ + e⁻ ⇌ COH or CHO. The potential dependent barrier for this step has been examined by several groups. It would be helpful if this barrier or free energy of reaction was calculated and added to the discussion.

(3) They note in the SI that they examined CO dimerization on both (111) and (100) facets for the Cu-on-Cu₂O and Cu-on-Cu₃N. It would be helpful to report all the various surfaces that were examined and the results in Table S1. If only one facet is showing preference for CO dimerization it would be useful to know this since presumably one could explore how that correlates to the morphology of the experimental catalyst nanoparticles.

Because of the promising experimental results and the difficulty of resolving a clear mechanism for this complex reaction/catalyst, I do not believe the DFT has to provide a irrefutable explanation of the results (there is ongoing debate of these questions in the literature and it is fair to say that this is outside of the scope of this paper) but it would be helpful if they provide context to what has been done in the literature and are clear on their calculations. All of these additions could be made with a more detailed description, with references, of the DFT calculations in the SI.

I would recommend that the paper be accepted once the details suggested above are added to the SI.

Response and list of changes implemented

“Copper-on-nitride enhances the stable electrosynthesis of multi-carbon products from CO₂”

Reviewer #1 (Remarks to the Author):

Contents of manuscript: Authors prepared Cu on Cu₃N, which they believe to be an effective catalyst for producing C₂ products from CO₂. The premise for this move is that the Cu₃N core helps to keep the outer layer of Cu oxidized. The highest CO₂R activities found are at -0.95 V RHE: C₂H₄ (39% FE), C₂H₅OH (19% FE) and propanol (6% FE).

There are some issues with this manuscript:

1. No operando or in situ characterization techniques were done to ensure that the Cu outer layer stays oxidized during CO₂ reduction. Hence, the foundational evidence to support the basic claims of this manuscript is not there.

As the reviewer recommends, we now provide *in-situ* X-ray absorption spectroscopy (XAS). We investigate thereby the effect of the Cu⁺ support on the structure and chemical state of the catalyst, all as a function of time under operation.

Fig. R1 shows the Cu K-edge spectra of Cu-on-Cu₃N catalyst at -0.95 V vs RHE during CO₂ reduction. XAS of the as-prepared catalyst (0 min) indicates an absorption edge between Cu (8979 eV) and Cu₃N (8980.5 eV) – and in particular exhibits a prominent shoulder at 8980.0 eV. Over the course of CO₂ reduction, both Cu and Cu₃N features are still present (Fig. R1a-b), with a shoulder energy at 8979.4 eV after two hours. In contrast, the Cu-on-Cu₂O catalyst shows a prominent metallic Cu feature after one hour under CO₂ reduction (Fig. S10a).

To gain more insight into the role of the Cu⁺ support, we recorded *in-situ* Cu K-edge spectra of Cu-on-Cu₃N and Cu-on-Cu₂O catalysts following 30 min under CO₂ reduction (Fig. R1c). We found that the absorption edges of the two catalysts are between Cu⁺ and Cu⁰, indicating the presence of a mixture during the reaction. However, the absorption edge of Cu-on-Cu₂O was at a lower energy than that of Cu-on-Cu₃N, with energies at 8979.4 eV and 8979.8 eV, respectively. We also calculated the ratio of Cu oxidation states with respect to the reaction time (Fig. R1d). The Cu-on-Cu₃N catalyst shows that the structure becomes stable with Cu₃N and Cu after the initial 60 min, while Cu-on-Cu₂O presents only the Cu component after one hour. This observation indicates suppression of the partial reduction of the catalyst when we use the Cu₃N support.

Fig. R1. *In-situ* characterization of the structure and chemical state for the catalysts during CO₂ reduction. **a**, *In-situ* Cu K-edge XAS spectra of Cu-on-Cu₃N catalyst as function of reaction time at -0.95 V vs RHE. **b**, The first derivatives of the XAS spectra in **a**. **c**, *In-situ* Cu K-edge XAS spectra during the first 30 min on the catalysts: Cu-on-Cu₃N (green) and Cu-on-Cu₂O (orange). Spectra of Cu (wine red) and Cu₃N (earthy yellow) are also listed as references. **d**, Calculated ratio of Cu⁺ species with respect to the reaction time at -0.95 V vs RHE.

2. The CO₂R activities reported above are supposed to be the best. But really, they are not any better than many previous works using the so-called Cu₂O-derived Cu.

In this work, we aimed at improving CO₂ reduction stability while achieving a high selectivity and activity comparable to reported Cu₂O-derived catalysts.

Prior studies have found that the mixture of Cu⁺/Cu⁰ active species synergistically promotes CO₂ reduction in favour of C₂₊ selectivity. However, whether there exists a stable presence of active Cu⁺ during CO₂ reduction has remained the subject of debate.

Previous results suggest that metal nitrides offer a stable compound and can be employed not only as the catalytic active species but also as support¹. In our report, we present a Cu-on-Cu₃N catalyst in which the surface Cu protects the Cu₃N support from further reduction over the course of CO₂ reduction, a fact we now confirm in detail using *in-situ* XAS results (Fig. 3a-b in the revised manuscript). This composite leads to a modified Cu electronic structure affected by the Cu₃N support. Together with the suppressed reduction of Cu₃N support, the new strategy results in stable selectivity for at least 30 hours with only 10% of the loss in CO₂ reduction selectivity.

We summarize the results and compare the present advance to the best prior reports in Table R1.

Table R1. Comparison of the performance in CO₂ reduction to C₂₊ products using Cu-based catalysts.

Catalyst	FE toward C ₂₊ products (%)	J C ₂₊ partial current density (mA/cm ²)	Stability (h)	Ref.
Cu ₂ O derived Cu	55	17	1	ACS Catal. 2015, 5 , 2814
Cu nanocubes	48	1	1	Angew. Chem. Int. Ed. 2016, 55 , 5789
Cu _x Zn	51	14	1	ACS Catal. 2016, 6 , 8239
Cu ₂ O derived Cu	59	18	1.2	J. Phys. Chem. C , 2016, 120 , 20058
Plasma-activated Cu	60	7	5	Nat. Comm. 2016, 7 , 12123
Plasma-activated Cu nanocubes	71	25	Not reported	ACS Nano , 2017, 11 , 4825
Cu	63	2	Not reported	PNAS , 2017, 114 , 5918
Cu nanoparticles	55	11	10	PNAS , 2017, 114 , 10560
Cu-on-Cu₃N	64	14	32	This work

3. Minor: Please include error bars in data.

We repeated each experiment three times, and took the average as the final value. We added the error bars in Fig. 4a-d and Fig. S12.

4. The authors has from the very start, make the statement that Cu⁺ are active sites, and has to be protected from reduction. This claim is not that well supported in the CO₂R community. Could the authors give a more balanced viewpoint that metallic Cu sites with defects could also work for CO₂R -> C₂₊. Moreover, how about the roles of local pH, as proposed by Koper's group?

Inspired by the reviewer's suggestion, we reviewed the proposed reaction mechanism in light of the experimental results. We now provide an improved explanation in the manuscript (pp. 9-11).

We first evaluated the effect of surface defects on C_{2+} selectivity. For Cu-on- Cu_2O catalyst, we believe that surface defects – such as grain boundaries – may play a role in the selectivity toward C_{2+} in the case of the oxide-derived process. For the Cu-on- Cu_3N catalyst, surface defects can also affect C_{2+} selectivity. However, compared with the Cu-on- Cu_2O , which was quickly reduced to Cu (Figure 3d in the revised manuscript), the Cu-on- Cu_3N catalyst retained a higher C_{2+} selectivity under CO_2 reduction. We offer that suppressed reduction of the Cu_3N support thus play a significant role in the high selectivity over the course of CO_2 reduction. Together with the prolonged presence of Cu^+ over time, it allows for increased-stability C_{2+} electrosynthesis under CO_2 reduction.

We also considered local pH as another possible contributor. It is expected that local pH will rise on rough surfaces, suppressing pH-dependent CO protonation toward methane formation; while the pH-independent CO dimerization is unaffected, resulting in increased C_{2+} selectivity²⁻⁶.

Compared to the pure Cu catalyst, Cu-on- Cu^+ catalysts display a suppression in methane selectivity, which can be attributed to increased local pH. We also performed CO_2 reduction in buffer solution (0.1 M K_2HPO_4 + 0.1 M KH_2PO_4 , pH = 6.8). The results show that the Faradaic efficiency of methane increases while ethylene decreases for the Cu-on- Cu^+ catalysts compared to that in 0.1 M $KHCO_3$ (Fig. R2), which confirms the effect of local pH. Comparing Cu-on- Cu_2O and Cu-on- Cu_3N catalysts, the geometric current densities are similar (Fig. S15), which indicates a nearly identical consumption rate of local protons during CO_2 reduction. We propose therefore that differences in local pH do not account for the higher C_{2+} selectivity for Cu-on- Cu_3N relative to Cu-on- Cu_2O .

As demonstrated by the *in-situ* XAS and STEM-EELS results (Fig. 2-3 in the revised manuscript and Fig. S6), both Cu and Cu_3N features are observed during CO_2 reduction. EELS mapping revealed that the Cu surface is no more uniformly-shaped following the electroreduction (Fig. S6). We conclude that Cu^+ species are supplied to subsurface layer during the reduction, resulting in the selectivity for C_{2+} . DFT calculations reveal that Cu_3N support modifies the electronic structure of the surface Cu, decreasing the energy barrier of CO dimerization (Fig. S23). Together with the suppressed reduction of Cu^+ using Cu_3N as a support, we achieve the highest C_{2+} production under CO_2 reduction among the three catalysts.

Fig. R2. Product distribution for CH₄, C₂H₄, CO and H₂ on the three catalysts at -0.95 V vs RHE in CO₂-saturated buffer solution (pH = 6.8).

Because of # 1, #2 and #4, I am not able to recommend acceptance of this manuscript, in its present form. Much more work needs to be done, especially to address #1.

We performed each of the experiments proposed by the reviewer, including *in-situ* XAS and STEM-EELS to examine the catalyst evolution during CO₂ reduction, and the relevant electrochemical measurement in buffer solution to exclude the effect of local pH. We also carried out DFT calculations to investigate the relationship between the catalyst structure and the catalytic performance. Based on these results, we now provide improved substantiation of the proposed mechanism within the revised manuscript.

Reviewer #2 (Remarks to the Author):

This work is a combined theory and experiment study that examines the effect of oxidized Cu supports on the activity, selectivity, and stability of thin Cu overlayers. Overall, the authors conclude that a lower kinetic barrier for C-C coupling in the case of Cu overlayers on both Cu₃N and Cu₂O supports leads to an increase in activity for CO₂ reduction to C₂+ products. The impact of sub-surface oxygen on the catalytic behavior of Cu-based catalysts has been a widely-debated topic in the CO₂ electrochemistry field, and developing a better fundamental understanding on the structure and stability of active-site motifs in such composite catalysts is of general interest to the scientific community. While the performance of the composite Cu-based catalysts in this study are very impressive, the correlation between structure and activity is weak and will likely require additional experiments to substantiate the conclusions made by the authors. As such, I cannot recommend that this work is published in Nature Communications in its current state, and have outlined my concerns below.

1. According to a recent theoretical study by Garza, Bell, and Head-Gordon, the diffusion rates of anions such as oxygen in an oxidized Cu lattice are expected to be quite fast, and thus, sub-surface Cu oxides are expected to have poor stability within the atomically thin overlayer structures suggested by the authors (DOI: 10.1021/acs.jpcclett.7b03180). The partially reduced Cu₂O and Cu₃N structures generated by DFT have oxygen or nitrogen within the first several monolayers of the surface. In the cases where metallic Cu is an overlayer on Cu₂O or Cu₃N, what is the expected stability of the oxidized structures in either the 3rd or 2nd monolayer, respectively?

In light of the reviewer's suggestion, we built different numbers of Cu layers on top of the Cu₃N and Cu₂O supports (Fig. S18-21) and calculated the diffusion free energy barriers of the oxygen and nitrogen atoms from core to the surface. Our calculations show that there is a large diffusion barrier (more than 2 eV) for both O and N to diffuse from their original positions within Cu₂O and Cu₃N, respectively, to one layer above their original position (diffusion from level 0 to level 1, Fig. S24-26). This is in contrast to the report of Garza *et al.*⁷; however, in the Garza model, the oxygen in the sublayer is a defect within pure copper. In contrast, in our model, nitrogen/oxygen are within the pristine crystalline structure of Cu₃N or Cu₂O, and their diffusion to the top Cu layers will not only will leave a vacancy defect within Cu₃N and Cu₂O structures, but will also create another defect in the top pure Cu layers. We now explain that larger energy barriers for oxygen and nitrogen diffusion are reasonable in our catalyst model compared to the sublayer oxygen model studied by Garza *et al.*

We also studied the diffusion energy barriers from the first layer to the upper layers toward the surface (up to level 5). Interestingly, as in Garza's study, we observed no energy barrier for oxygen diffusion at these stages. However, there was another energy barrier for nitrogen diffusion from subsurface layers (level 3 and 4) to the surface (level 5) of around 1 eV. This energy barrier leads nitrogen to be trapped in the sublayers near the surface (all initial and relaxed configurations are in Fig. S24-26 and all relevant energy barriers are summarized in Table S6.).

Therefore, based on our computational results, we posit that, oxygen diffusion is easier than nitrogen diffusion. As a result, the Cu-on-Cu₂O will convert to pure Cu much faster than Cu-on-Cu₃N. For Cu-on-Cu₃N, we expect to have nitrogen on sublayers close to the surface, and to be able to observe its effect on the surface copper's electronic structure. In addition, having a thick Cu layer on top of both Cu-on-Cu₂O

and Cu-on-Cu₃N will introduce a large barrier for both oxygen and nitrogen to diffuse from the pristine core to the pure copper shell, and this delays reduction of these compounds to pure copper.

In addition, experimentally, EELS mapping shows a Cu₃N signal after the reaction (Fig. S6). Due to the change of the surface structure, the Cu outlayer is no longer uniformly shaped, and some of Cu⁺ species may be supplied to the subsurface layer (Fig. S6f), resulting in a selectivity for C₂₊ during the reduction. It is likely that these Cu⁺ species remain relatively stable in the sublayer since the activity and selectivity remain constant within the reaction time. Furthermore, the *in-situ* XAS data also show the presence of Cu and Cu₃N during the 120 min and a relatively stable ratio of Cu₃N/Cu after one-hour of reaction (Fig. 3 in the revised manuscript).

2. The geometric arrangement of Cu atoms within the metallic overlayers on Cu₂O and Cu₃N appear to be quite different. Previous work in single crystal electrochemistry has shown that the C-C coupling activity of Cu electrodes are quite sensitive to the exposed facet. Are these the most thermodynamically stable facet terminations for the metallic overlayer? Otherwise, why were these particular facets chosen? There doesn't seem to be a strong correlation between the CO adsorption energies and the kinetic barriers for CO dimerization. How much of the decrease in the kinetic barrier for CO dimerization is due to a geometric effect vs. a ligand or strain effect from the support?

To investigate geometric effects, we compared different catalysts with the same facets. We considered the two widely-studied (111) and (100) facets as the most stable closed pack surface and active surface for C-C coupling, respectively, for the Cu-on-Cu₂O and Cu-on-Cu₃N catalysts as well as Cu, Cu₂O and Cu₃N.

Our calculations show that, as in previous studies, (100) shows a stronger catalytic activity toward CO dimerization compared to the (111) facet. However, comparing different models with the same facet, we observe the ligand effect and the effect of the altered electronic structure due to the sublayer nitrogen atoms. These models are present in Fig. S17-21, and the calculated energy barriers for CO dimerization are listed in Table S5.

3. In the best case scenarios of having the metallic overlayers on Cu₂O and Cu₃N, the improvements in CO dimerization barrier height appear to be quite small in comparison to metallic Cu (~0.1 eV). In particular, the differences in barrier height between the metallic overlayers on Cu₂O and Cu₃N appear to be very small. Are these improvements within the error of the DFT calculations?

We agree that the difference in thermodynamic energy barriers is quite small. However, the activation energy barrier calculated from the transition state is larger (0.412 eV). We added these calculations to the Supplementary Computational Details portion (pp. 4-8) and the calculated activation energy barriers are listed in Fig. S23. We considered Cu, Cu-on-Cu₂O (with 1 Cu top layer), and Cu-on-Cu₃N (with 1 Cu top layer) with (100) facet. We expect that the same trend will be observed for the other models studied in this paper.

4. For the Cu₂O or Cu₃N support to affect the binding of reaction intermediates through either ligand or strain effects, the metallic Cu overlayer must be very thin and the authors suggest a thickness of ~1-2 monolayers. However, the combination of physical characterization techniques utilized by the authors does not have a high enough spatial resolution to determine how thick this metallic Cu overlayer is on the Cu₂O or Cu₃N support. Experimentally demonstrating either the thickness of the overlayer or that its electronic structure is affected by the Cu₂O or Cu₃N support is critical to substantiate the enhancement mechanism that the authors have proposed. Have the authors considered valence band spectroscopy measurements to examine the differences in electronic structure between the metallic Cu nanoparticles and composite structures?

As recommended by the reviewer, we re-examined the HRTEM-EELS mapping of Cu-on-Cu₃N catalyst before (Fig. R1a-c) and after two-hour reaction (Fig. R1d-f). The catalyst surface exhibited indications of surface reconstruction following operation under reduction conditions.⁸ As shown in Fig. R1c and R1f, our analysis indicates a maximum 3 nm of the surface Cu layer, while in some areas, the thickness is less than 1 nm. It is likely that some Cu⁺ species will reside in the subsurface layer during reduction, resulting in selectivity for C₂₊.

Fig. R1. TEM characterization of the Cu-on-Cu₃N catalyst before and after two-hour reaction. a and d, HADDF-STEM images before (a) and after (d) reaction. b-c, STEM-EELS Cu and N element mapping of the particle in a. e-f, Cu and N element mapping of the particle in d.

We also measured the valence band spectra (VBS) to examine the difference in the valence electronic structure between the Cu and the composite (Fig. R2). Compared with pure Cu, we found that the Fermi-level (E_F) shifted toward VB_m by 0.33 eV for Cu-on-Cu₃N and 0.08 eV for Cu-on-Cu₂O, respectively, indicating that the core-level Cu₃N or Cu₂O support affect the electronic structure of the surface Cu.

Fig. R2. XPS valence band spectra at the surface on the three catalysts: Cu-on-Cu₃N catalyst (green dots); Cu-on-Cu₂O catalyst (purple dots); and pure Cu (orange dots).

We also considered systems having different Cu monolayers on top of Cu₂O and Cu₃N. As the reviewer mentioned, the effect of sublayer nitrogen and oxygen on the surface electronic structure dramatically decreases when one increases the number of copper overlayers (the electron localization potential of these models with different Cu layers are shown in Fig. S18-21, and the calculated charge density by the Bader charge analysis are tabulated in Table S5). However, our diffusion barrier calculations show that nitrogen atoms can exist on sublayers very close to the surface, and that it still has a significant effect on the surface electronic structure and consequently on the CO dimerization.

We now add data and related explanations to the revised manuscript and the Supplementary Materials (Fig. S6, S9, S18-21, Table S5).

5. Instead of a strain or ligand effect from the Cu₂O or Cu₃N support, have the authors considered a mechanism of enhancement from a difference in exposed Cu facets? Single-crystal studies have shown similar trends in C₂+/C₁ ratios through improvements in C-C coupling activity (and CH₄ suppression) from introducing more undercoordinated sites on the surface (DOI: 10.1007/978-0-387-49489-0_3).

The previous results show that single-crystal Cu (111) facets favor methane formation, while (100) facets help to increase the C-C coupling toward ethylene production⁹⁻¹².

In this work, we synthesized three catalysts using an initial electroreduction of the surface oxidation layer using a negative cyclic voltammetry (CV) scan, yielding the final catalyst. During this process, the Cu species possesses a polycrystalline structure (Fig. S16). These structures do not exhibit the specific facet

orientation. Therefore, we would not expect that they contribute in a quantitatively significant way to increase C_{2+} selectivity.

6. How does the stoichiometry of the Cu overlayer on Cu_2O or Cu_3N composite change after steady state electrocatalysis measurements? Are there any bulk structural changes in the catalyst after these longer-term measurements? Is there any evidence of O^{2-} or N^{3-} leaching over time?

We found that, following the steady-state electrocatalysis measurements, the catalyst began to lose activity and saw a loss in performance. We acquired TEM images of the Cu-on- Cu_3N catalyst following the long-term tests, and the results showed an aggregated morphology (Fig. R3). We were not able to determine the change of Cu stoichiometry of Cu the overlayer on this Cu_3N composite by this measurement.

Fig. R3. TEM characterization of the Cu-on- Cu_3N catalyst after the long-term electrocatalysis test.

We refer therefore to *in-situ* XAS results which show that the ratio of $Cu^+/(Cu^++Cu^0)$ decreased and then gradually became stable after the initial 60 min (Fig. 3d in the revised manuscript). Since the sample is stable after 60 min, we characterized structural changes following two hours. TEM images show that the bulk Cu_3N is still present, but is decreased compared with the case before reaction (Fig. R1). This is consistent with the *in-situ* XAS results.

Overall, we conclude, based on *in-situ* XAS and TEM mapping results, that the stoichiometry of the Cu overlayer on Cu_3N composite increases, the bulk structure is still present but decreased, and that N^{3-} leaches out as a result of the reaction.

For the Cu-on- Cu_2O catalyst, *in-situ* XAS results show a prominent metallic Cu feature after 60 min (Fig. S10). Therefore, we also believe that the Cu overlayer increases, bulk structure changes to derived Cu, and O^{2-} leaches out after the reaction compared with that before reaction.

We point out that our diffusion free energy barrier calculations support the view that the oxygen in sublayer is less stable than is the nitrogen one, consisting with our finding that Cu-on- Cu_3N exhibits increased stability compared to Cu_2O .

7. Why is the Cu overlayer on Cu₃N or Cu₂O more stable than pure Cu? The transmission electron micrographs in the Supporting Information don't show much change in the particle morphology after the reduction treatment. Have the authors used ICP measurements to determine whether there is any dissolution or delamination of Cu from the electrodes over time?

As recommended by the reviewer, we obtained ICP of the electrolyte at various reaction times of each catalyst (Fig. R4).

For pure Cu, we found that a certain content of Cu was delaminated into the electrolyte, which might cause the poor stability of CO₂ reduction. For the Cu-on-Cu₃N and Cu-on-Cu₂O catalysts, the concentrations of Cu dissolution are very low with time evolution. We conclude that Cu dissolution is expected to have a negligible effect on the stability.

Based on the VBS (Fig. S9), *in-situ* XAS (Fig. 3 in the revised manuscript) and STEM-EELS mapping (Fig. S6) results, we propose that the Cu₃N support affects the electronic structure of the surface Cu and thereby the performance of CO₂ reduction. Together with the suppressed reduction of Cu⁺ over time, which prolongs the effect of the Cu₃N support on the surface Cu, it allows for an increased C₂₊ electrosynthesis stability during CO₂ reduction.

Fig. R4. The concentration of the Cu dissolution into the electrolyte on the three catalysts: Cu-on-Cu₃N (red dots-line); Cu-on-Cu₂O (yellow dots-line); and the pure Cu (blue dots-line).

8. Please check the fitting of the X-ray photoelectron spectra. In particular, the fitting for the Cu Auger LMM spectra needs more attention to be published.

We have corrected the fitting curve of Cu Auger LMM spectra in Fig. 1 in the revised manuscript.

9. *It is difficult to deconvolute the effect of the ligand from the catalytic behavior of Cu₃N for the measurements shown in Figure S13 and Table S6.*

We agree with reviewer that the ligand affects the reactivity of Cu₃N catalyst. CO₂ reduction shows a low activity because of the poor conductivity of Cu₃N capped with surface long-chain organic ligand. We also confirm that all reduction products come from CO₂ rather than the ligands – a fact we validated using N₂-saturated electrolyte in control experiments (Fig. S13).

To avoid confusing the reviewer and readers, we have removed this dataset from the revised manuscript since it secondary to the main narrative.

Reviewer #3 (Remarks to the Author):

This study explores the design of Cu-based catalysts where Cu metal (Cu⁰) is deposited on a Cu+ support to enhance the selectivity of CO₂ reduction to C₂+ products. The idea is partly supported by literature work on Cu oxide derived catalysts and DFT calculations within the paper evaluating CO adsorption energy versus CO dimerization barrier. The hypothesis is that the selectivity step between C₁ and C₂ products is hydrogenation of CO versus CO dimerization. Based on the finding that Cu on Cu₃N shows that the CO dimerization barrier is lowered in comparison to Cu metal and Cu on Cu₂O, they synthesis Cu-on-Cu₃N catalysts. These catalysts were tested along with Cu and Cu-on-Cu₂O catalysts. They see that the FE to C₂ is enhanced substantially on the Cu-on-Cu₃N catalysts. Very interestingly (and this is rare in the CO₂ electroreduction work in the literature) they report the stability of the CO₂ electroreduction over 32 hours and see much more stability than Cu-on-Cu₂O and Cu metal.

This stability result is promising since one of the issues with much of the CO₂ electroreduction work is that the performance of the catalysts degrades rapidly.

Overall, this paper reports a very interesting new finding for CO₂ electroreduction. This is a hot area with a lot of results coming quickly in the past 2-3 years but this paper is promising since they not only see a dramatic increase in selectivity but also study stability and have a hypothesis-driven approach to the catalyst design. I am not entirely convinced their hypothesis is correct for the reasons they give – i.e enhance CO dimerization. The main suggestion and question I have are related to the DFT calculations, which I found to be relatively weak component of this paper.

We appreciate the reviewer's feedback. We have acted on each suggestion as documented below, with a major focus on improving the DFT studies in the manuscript. We add detailed information to the revised manuscript (pp. 11-12) and the Supplementary Materials (pp. 4-8, Fig. S17-26, Table S5-6).

1. They should report more detail on the CO dimerization calculation. There has been some controversy about this pathway – the initial proposal was that a CO + CO⁻ reaction occurs. Not two neutral CO dimers, which was found to have a very large barrier. The more recent work from the Norskov group examines this reaction and showed that cations and protons could be used to facilitate this reaction. There are no references to the work of Koper and Norskov (and also Goddard on the CO dimerization on Cu(100)) in this paper. I realize that references are limited but the authors should provide this context in the SI and explain what was involved in their CO dimerization calculation in reference to this work. If two neutral CO molecules were coupled to form a CO dimer then this would be a non-electrochemical step and would not explain any of the potential dependence seen in the experiments.

In the revised Supplementary Materials, we now discuss prior mechanistic studies on CO dimerization by the Koper, Norskov, Goddard, Bell, Head-Gordon, Janik and Asthagiri groups. Our methodology is based on the non-electrochemical CO dimerization to evaluate potential-independent catalytic activity of three systems: Cu, Cu-on-Cu₃N and Cu-on-Cu₂O. We recognize that considering the parameters involved in the reaction such as anions and cations, applied potential and also surface coverage of the reactants will improve the quality of the DFT computations and will provide a more realistic model. Therefore, as in

previous works from the above-mentioned groups, we considered the CO dimerization on different models to see the effect of the structure and consequently the oxidation state on the catalytic activity.

2. If the hypothesis is that C1 versus C2 is a competition between CO hydrogenation and CO dimerization, does it not make sense to look at the barrier to CO hydrogenation on these various surfaces? i.e. $CO^ + H^+ + e^- \rightarrow COH$ or CHO . The potential dependent barrier for this step has been examined by several groups. It would be helpful if this barrier or free energy of reaction was calculated and added to the discussion.*

In light of the reviewer's suggestion, we calculated the reaction free energy barrier for CO hydrogenation (both COH and CHO) on both Cu-on-Cu₃N and Cu-on-Cu₂O catalysts with different number of Cu layers on top (Table S5). Our computations show that the energy barrier for CO hydrogenation on Cu-on-Cu₃N (0.933 eV) is higher than that on Cu-on-Cu₂O (0.749 eV) and pure Cu (0.721 eV) on (100) facets, indicating improved catalytic activity toward C₂₊ production on Cu-on-Cu₃N catalyst.

3. They note in the SI that they examined CO dimerization on both (111) and (100) facets for the Cu-on-Cu₂O and Cu-on-Cu₃N. It would be helpful to report all the various surfaces that were examined and the results in Table S1. If only one facet is showing preference for CO dimerization it would be useful to know this since presumably one could explore how that correlates to the morphology of the experimental catalyst nanoparticles.

We have calculated the CO adsorption energy and CO dimerization energy barrier for two different facets of (111) and (100) for both Cu-on-Cu₃N and Cu-on-Cu₂O catalysts with different number of Cu layers on top (Fig. S17-21 and Table S5). The relevant discussion of geometrical effects and oxidation state effects are also added to the Supplementary computational details (pp. 4-8).

Reference

1. Giordano, C. & Antonietti, M. Synthesis of crystalline metal nitride and metal carbide nanostructures by sol-gel chemistry. *Nano Today* **6**, 366-380 (2011).
2. Varela, A.S., Kroschel, M., Reier, T. & Strasser, P. Controlling the selectivity of CO₂ electroreduction on copper: the effect of the electrolyte concentration and the importance of the local pH. *Catal. Today* **260**, 8-13 (2016).
3. Kortlever, R., Shen, J., Schouten, K.J.P., Calle-Vallejo, F. & Koper, M.T. Catalysts and reaction pathways for the electrochemical reduction of carbon dioxide. *The journal of physical chemistry letters* **6**, 4073-4082 (2015).
4. Nie, X., Esopi, M.R., Janik, M.J. & Asthagiri, A. Selectivity of CO₂ reduction on copper electrodes: the role of the kinetics of elementary steps. *Angew. Chem. Int. Ed.* **52**, 2459-2462 (2013).
5. Calle - Vallejo, F. & Koper, M. Theoretical considerations on the electroreduction of CO to C₂ species on Cu (100) electrodes. *Angew. Chem.* **125**, 7423-7426 (2013).
6. Montoya, J.H., Shi, C., Chan, K. & Nørskov, J.K. Theoretical insights into a CO dimerization mechanism in CO₂ electroreduction. *The journal of physical chemistry letters* **6**, 2032-2037 (2015).
7. Garza, A., Bell, A.T. & Head-Gordon, M. Is Subsurface Oxygen Necessary for the Electrochemical Reduction of CO₂ on Copper? *The journal of physical chemistry letters* (2018).
8. Manthiram, K., Beberwyck, B.J. & Alivisatos, A.P. Enhanced electrochemical methanation of carbon dioxide with a dispersible nanoscale copper catalyst. *J. Am. Chem. Soc.* **136**, 13319-13325 (2014).
9. Hori, Y.i. in *Modern aspects of electrochemistry* 89-189 (Springer, 2008).
10. Schouten, K.J.P., Qin, Z., Pérez Gallent, E. & Koper, M.T. Two pathways for the formation of ethylene in CO reduction on single-crystal copper electrodes. *J. Am. Chem. Soc.* **134**, 9864-9867 (2012).
11. Hori, Y., Takahashi, I., Koga, O. & Hoshi, N. Selective formation of C₂ compounds from electrochemical reduction of CO₂ at a series of copper single crystal electrodes. *The Journal of Physical Chemistry B* **106**, 15-17 (2002).
12. Durand, W.J., Peterson, A.A., Studt, F., Abild-Pedersen, F. & Nørskov, J.K. Structure effects on the energetics of the electrochemical reduction of CO₂ by copper surfaces. *Surf. Sci.* **605**, 1354-1359 (2011).

Reviewers' comments:

Reviewer #1 (Remarks to the Author):

The authors have substantially revise the manuscript.

I agree with most of the changes made.

Regarding in situ expts to show that surface Cu atoms remained oxidized because of the underlying CuN (Reviewer 1, Q1): the authors used XAS. I do not think that this is a useful technique, as XAS probes into the bulk, while we are interested in the oxidation states of the surface Cu atoms.

Reviewer #2 (Remarks to the Author):

In their rebuttal, the authors have put together an extensive and scientifically sound response to the concerns raised by myself and the other reviewers. Major improvements include:

- A reconstructed introduction that better motivates the research topic.
- Additional stability and reaction barrier calculations, to provide more evidence for the mechanism of activity and selectivity enhancement.
- Better evidence for the Cu on Cu₃N structure with further HRTEM-EELS analysis, giving credence to the overlayer structure proposed by theory. The morphology is now supported by valence band spectroscopy that demonstrates a change in the electronic structure of Cu.
- XAS analysis, further supporting the existence of an oxidized Cu species in situ.
- Provided a more balanced discussion on possible mechanisms of performance enhancement.

As such, I believe that this work is now ready to be published in Nature Communications.

Reviewer #3 (Remarks to the Author):

The authors have added more support for the in situ oxidation state of the catalysts over time and these details support their hypothesis that they have protected Cu⁺ more in the Cu nitride based catalysts. The addition of the CO hydrogenation barrier and more clearer presentation of the DFT calculations also improves the paper. I would recommend the paper for publication.

Response and list of changes implemented

“Copper-on-nitride enhances the stable electrosynthesis of multi-carbon products from CO₂”

Reviewer #1 (Remarks to the Author):

The authors have substantially revise the manuscript.

I agree with most of the changes made.

Regarding in situ expts to show that surface Cu atoms remained oxidized because of the underlying CuN (Reviewer 1, Q1): the authors used XAS. I do not think that this is a useful technique, as XAS probes into the bulk, while we are interested in the oxidation states of the surface Cu atoms.

In light of the reviewer’s feedback, we sought a surface-sensitive method to further characterize the surface chemical states of the catalyst.

We now employ angle-resolved XPS (ARXPS), which provides a depth sensitivity on the order of nanometers, achieved by varying the emission angle at which electrons are detected from the surface¹⁻³. The detection depth is reduced as the emission angle θ is decreased (Scheme R1).

Scheme R1. Geometry of XPS. a, General XPS. b, Angle-resolved XPS.

To gain increased surface sensitivity, we acquired ARXPS at 20° emission angle with respect to the sample surface (Figure R1a). The detection depth is lower than 2 nm.

Cu LMM spectra (Figure R1b-f, left column) indicate the presence of Cu⁺ and Cu⁰, and the N 1s spectra (Figure R1b-f, right column) are consistent with the spectrum of metal nitride⁴, indicating the presence of Cu₃N in the first ~ 2 nm of the surface over the course of CO₂ reduction. In the initial 60 min, Cu⁰ content increased and Cu₃N content decreased; thereafter, such as following a two-hour reaction, the

catalyst gradually reached a stable surface composition. These data are consistent with the *in-situ* XAS results (Figure 3d in the revised manuscript).

In summary, combining with ARXPS (Figure R1), *in-situ* XAS (Figure 3) and STEM-EELS (Supplementary Fig. 6), we now provide further evidence that subsurface Cu₃N affects the electronic structure of the surface layer, thereby decreasing the barrier energy associated with CO dimerization during CO₂ reduction.

We have added this result as Supplementary Fig. 11 to the revised Supplementary Information and the related explanation to the revised manuscript.

Figure R1. Angle-resolved XPS (ARXPS) characterization of the catalyst with time evolution under CO₂ reduction. **a**, The scheme of ARXPS analysis at 20° emission angle with respect to the sample surface. **b-f**, ARXPS results of Cu LMM (left line) and N 1s (right line) spectra obtained with different reaction time: **b**, 0 min, **c**, 30 min, **d**, 60 min, **e**, 120 min, and **f**, 300 min, respectively.

Reference

1. Cumpson, P.J. Angle-resolved XPS and AES: depth-resolution limits and a general comparison of properties of depth-profile reconstruction methods. *J. Electron. Spectrosc. Relat. Phenom.* **73**, 25-52 (1995).
2. Watts, J.F. & Wolstenholme, J. An introduction to surface analysis by XPS and AES. *An Introduction to Surface Analysis by XPS and AES*, by John F. Watts, John Wolstenholme, pp. 224. ISBN 0-470-84713-1. Wiley-VCH, May 2003., 224 (2003).
3. Suzuki, S., Ishikawa, Y., Isshiki, M. & Waseda, Y. Native oxide layers formed on the surface of ultra high-purity iron and copper investigated by angle resolved XPS. *Mater. Trans., JIM* **38**, 1004-1009 (1997).
4. Wang, D. & Li, Y. Controllable synthesis of Cu-based nanocrystals in ODA solvent. *Chem. Commun.* **47**, 3604-3606 (2011).

REVIEWERS' COMMENTS:

Reviewer #1 (Remarks to the Author):

The authors have performed new AP-XPS (alongside XAS measurements performed earlier) to support their claim.

I do not think that their methodology is correct. This is because XPS is performed with an electrode out of the electrochemical cell. The electrode would thus have sufficient time to oxidize, etc. before the XPS measurement is done. Hence, what is measured cannot be representative of what is on the surface during electrochemical CO₂ reduction. And we all know that Cu oxidizes very quickly!

My suggestion is that the authors could point out in their manuscript the limitations of their measurements, i.e., keep open the possibility that there could be no Cu⁺ present on the surface during CO₂RR. I do not think that this will affect the quality of the manuscript. In fact, being transparent about limitations of one's measurements is always welcomed in this world where a lot of claims are hyped-up!

Response and list of changes implemented

Manuscript ID: NCOMMS-18-03655C

“Copper-on-nitride enhances the stable electrosynthesis of multi-carbon products from CO₂”

Reviewer #1 (Remarks to the Author):

The authors have performed new AP-XPS (alongside XAS measurements performed earlier) to support their claim.

I do not think that their methodology is correct. This is because XPS is performed with an electrode out of the electrochemical cell. The electrode would thus have sufficient time to oxidize, etc. before the XPS measurement is done. Hence, what is measured cannot be representative of what is on the surface during electrochemical CO₂ reduction. And we all know that Cu oxidizes very quickly!

My suggestion is that the authors could point out in their manuscript the limitations of their measurements, i.e., keep open the possibility that there could be no Cu⁺ present on the surface during CO₂RR. I do not think that this will affect the quality of the manuscript. In fact, being transparent about limitations of one's measurements is always welcomed in this world where a lot of claims are hyped-up!

We agree with the reviewer's comment that the oxidation state of Cu could be affected by exposing the samples to air during the ARXPS sample preparation. For this reason we also monitored the N signal in ARXPS as an indicator of the presence of Cu₃N (Supplementary Fig. 11, the N signal is coincident with the spectrum of metal nitride), and thus as an indicator of the presence of Cu⁺.

We also accept the reviewer's point regarding the limitations of ARXPS and XAS in evaluating Cu⁺ at the surface. As suggested by the reviewer, we have added the paragraph below to the revised manuscript (pp. 9):

“Both in-situ XAS and ex-situ ARXPS suggest the presence of Cu⁺ following CO₂ reduction. Further, the N signal suggests the presence of Cu₃N, and STEM-EELS mapping (Supplementary Fig. 6) shows evidence of Cu₃N in the subsurface layer following CO₂ reduction. Nevertheless, we point out also that XAS has a bulk penetration depth; and that air-sensitive Cu complicates the interpretation of the ARXPS studies herein. For these reasons, direct and unambiguous confirmation of the presence of Cu⁺ at the surface of the catalyst remains an ongoing opportunity for further advances in the field of Cu-based electrocatalysis and model development.”